# High-Throughput Screening of PAM-Flexible Cas9 Variants for Expanded Genome Editing in the Silkworm (*Bombyx mori*)

**DOI:** 10.3390/insects15040241

**Published:** 2024-03-30

**Authors:** Le Sun, Tong Zhang, Xinhui Lan, Na Zhang, Ruolin Wang, Sanyuan Ma, Ping Zhao, Qingyou Xia

**Affiliations:** 1Integrative Science Center of Germplasm Creation in Western China (Chongqing) Science City, Biological Science Research Center, Southwest University, Chongqing 400715, China; damon9452@email.swu.edu.cn (L.S.);; 2Key Laboratory for Germplasm Creation in Upper Reaches of the Yangtze River, Ministry of Agriculture and Rural Affairs, Chongqing 400715, China; 3Engineering Laboratory of Sericultural and Functional Genome and Biotechnology, Development and Reform Commission, Chongqing 400715, China

**Keywords:** *Bombyx mori*, genome editing, protospacer adjacent motif, Cas9 variants, high-throughput rapid screening platform

## Abstract

**Simple Summary:**

Cas9 is limited by protospacer adjacent motif (PAM) sequences during the application process, which is particularly evident in insect genome editing. Researchers have discovered Cas9 variants with a wider range of recognition due to the development of genome-editing technology. However, the application of these variants in insects has been ineffective and time-consuming. To tackle this issue, this study established a high-throughput assay system capable of rapidly detecting recognizable PAM sequences of different Cas9 variants in *Bombyx mori* and the corresponding editing efficiency of the sites. We utilized this system to evaluate the editing ability of two protein variants, xCas9 and Cas9-NG, in *Bombyx mori* cells. Based on the editing ability of these two proteins, the editable range of the *Bombyx mori* genome can be greatly expanded, accelerating research on the genome’s function. This system is also applicable to other insects for detecting the editing ability of Cas9 variants, thus accelerating the study of silkworm genome function.

**Abstract:**

Genome editing provides novel opportunities for the precise genome engineering of diverse organisms. Significant progress has been made in the development of genome-editing tools for *Bombyx mori* (*B. mori*) in recent years. Among these, CRISPR/Cas9, which is currently the most commonly used system in lepidopteran insects, recognizes NGG protospacer adjacent motif (PAM) sequences within the target locus. However, Cas9 lacks the ability to target all gene loci in *B. mori*, indicating the need for Cas9 variants with a larger editing range. In this study, we developed a high-throughput screening platform to validate Cas9 variants at all possible recognizable and editable PAM sites for target sequences in *B. mori*. This platform enabled us to identify PAM sites that can be recognized by both xCas9 3.7 and SpCas9-NG variants in *B. mori* and to assess their editing efficiency. Cas9 shows PAM sites every 13 base pairs in the genome, whereas xCas9 3.7 and SpCas9-NG have an average distance of 3.4 and 3.6 base pairs, respectively, between two specific targeting sites. Combining the two Cas9 variants could significantly expand the targeting range of the genome, accelerate research on the *B. mori* genome, and extend the high-throughput rapid screening platform to other insects, particularly those lacking suitable NGG PAM sequences.

## 1. Introduction

CRISPR/Cas is an RNA-based immune system found in numerous archaea and bacteria that counteracts invasion through foreign genetic elements [1,2,3,4]. Many different types of CRISPR/Cas systems are present in nature, among which is the type II CRISPR/Cas9 system, which has been extensively utilized in research areas such as genome editing, gene regulation, and base editing. Mediated by this technique, genome editing has been utilized to quickly and effectively adjust endogenous genes in several insect species. The CRISPR/Cas9 system has been used to study gene function and pathways in various insects, including Diptera [5,6], Coleoptera [7], Hymenoptera [8], Lepidoptera [9,10,11], Hemiptera [12], and Orthoptera [13].

As a model insect in the order Lepidoptera, genome-editing tools for *Bombyx mori* (*B. mori)* have also progressed, and with tools such as zinc finger nucleases [14], transcriptional activator-like effector nucleases [15], and CRISPR/Cas9 [11], precise genome-editing has been achieved in *B. mori* over the past five years [16,17,18]. Several novel mechanisms have been uncovered using genome-editing tools. Daimon et al. revised the long-accepted paradigm of JH signaling by generating and characterizing knockout *B. mori* with null mutations in JH biosynthesis or JH receptor genes using genome-editing tools [19]. Moreover, Zhu et al. systematically investigated the non-homologous end joining (NHEJ) DNA repair pathways and increased homologous recombination efficiency [20]. Many long-sought ideas, such as bioreactors with highly efficient silk glands, have also been achieved [21]. Furthermore, comprehensive genome-editing studies and their multiplex applications have promoted the development of biotechnology, including the first demonstration of heritable large chromosomal segment deletion [15] and the invention of novel microhomology-mediated, end-joining-dependent gene-insertion methods [22].

Engineered versions of CRISPR/Cas9 usually contain a Cas9 protein and a single guide RNA (sgRNA), with the DNA expected to correspond to the first 20 base pairs of the sgRNA (referred to as the “spacer sequence”), and have specific protospacer adjacent motif (PAM) sequences flanking the spacer sequence [23,24,25]. The existence of PAM sequences enhances the accuracy of the Cas9 editing system while simultaneously constraining its applicability, particularly given the emergence of more advanced editing approaches such as base editing. The Cas9-based CRISPR editing system in the silkworm family can only target 82.66% of genes and 79.87% of promoter regions, focusing solely on theoretically editable loci [26]. Considering various issues, such as ineffective sgRNAs and the difficulty of editing some loci, this number would decrease even further. As research advances, additional editing methods, including saturation and base editing, will be required to conduct more comprehensive studies. Both methods require Cas9 variants with a broader recognition range, particularly for base editing, which has a stringent editing window outside of which base editing is ineffective. Therefore, it is imperative to use Cas9 variants with large recognition ranges for successful base editing. Studies have revealed that the effective editing window for base editing in *B. mori* cells is within C4-C7 [27]. Therefore, the currently established CRISPR/Cas9 editing system is limited in its ability to perform precise site editing, restricting the process of gene-function research. Based on these limitations, it is imperative to identify proteins that can recognize a diverse array of PAM sites to broaden the scope of genome editing in *B. mori*.

The study of mammals faces a similar problem that must be resolved. To address this issue, scientists have pursued two research directions. The first direction involves the screening of several strains to obtain Cas9 proteins with varied editing abilities. To date, various Cas9 proteins capable of recognizing non-specific PAM sequences have been screened, including but not limited to CjCas9 [28], *Lachnospiraceae bacterium ND2006* Cpf1 [29], *Staphylococcus aureus* SaCas9 [30], and *Streptococcus canis* ScCas9 [31]. These proteins typically necessitate a distinct sgRNA scaffold or are too voluminous to introduce into host cells; therefore, they are not commonly used for this purpose. In addition to screening naturally occurring Cas9 proteins, researchers have explored the Cas9 protein. By modifying the amino acid composition of Cas9, attempts have been made to enable the protein to recognize non-specific PAM sequences. This has led to the discovery of Cas9 variants with altered specificity to NGG, NGA, NGNG, NGCG, and NAAG through both evolutionary and high-throughput screening [32,33,34]. Since 2018, scientists have developed mutant versions of Cas9 with extended PAM ranges, namely xCas9-3.7, SpCas9-NG targeting NGN, and SpCas9-NRRH/NRTH/NRCH targeting non-G PAMs [24,35,36,37]. More recently, two nearly PAM-less Cas9 mutants, SpRY and SpdNG-LWQT, were also created, covering almost all PAMs (NRN > NYN), though research shows their overall activity is reduced [38,39]. These Cas9 variants dramatically lowered PAM restriction, displaying the ability to modify PAM specificity via laboratory protein evolution. Previously, the Cas9 editing system was used to develop a genome-editing platform for *B. mori* that employed NGG as the PAM sequence. Genome analysis has revealed that although the majority of genes contain at least one target sequence, numerous genes still lack valid target sites [26], thereby limiting gene-function studies. Consequently, the development of new Cas9 variant proteins is urgently required to overcome this limitation. In this study, we established a high-throughput platform to detect Cas9 variants that overcomes the limitations of previous detection methods such as extended duration, high cost, and insufficient efficacy. This platform was used to evaluate the editing efficiencies of xCas9 3.7 and SpCas9-NG in BmE cells based on recognizable PAM sequences. xCas9 3.7 efficiently edited most unconventional PAM sequences, except for the CGN site, which showed lower efficiency in BmE cells. In contrast, SpCas9-NG successfully edited the target sequences with AGA, TGA, CGA, TGT, GGT, AGC, and TGC PAM sites. The utilization of these two Cas9 variant proteins has the potential to significantly expand the editing range of *B. mori*, thereby establishing a foundation for highly specific base editing. This detection method can serve as a reference for the establishment of other insect genome-editing systems. Additionally, this system can be applied to mammalian cells to quickly determine the PAM sequences recognized by new Cas9 variants and their corresponding editing efficiency.

## 2. Materials and Methods

### 2.1. Design and Construction of the Vectors

xCas9 3.7 and SpCas9-NG sequences were optimized for codon usage, synthesized by a commercial service (Genescript) (Tsingke, Beijing, China), and introduced into hr3-Hsp70-Cas9 by replacing the fragment encoding Cas9. Additionally, the hr3-Hsp70 promoter was replaced with IE2 to enhance Cas9 protein expression. Random 20-base sgRNA sequences were designed and utilized as targeting sequences, with target sequences containing any combination of three bases at the 3′ end as PAM sequences. The sgRNA scaffolds were added between the sgRNA and target sequences, and all target oligomers were synthesized, annealed, and ligated with AaRI-digested T-U6gRNA2. The plasmids are available upon request.

### 2.2. Cell Lines and Transfection

BmE is a cell line derived from *B. mori* embryonic tissues, maintained in our laboratory, and grown in Insect Grace medium at 27 °C supplemented with 10% fetal bovine serum (FBS). During sgRNA construction, pBac transposon arms were added, and with the help of A3helper, the sequence within the pBac transposon arms can be integrated into the cell genome. After transfection, the cells were cultured for one month to ensure the complete integration of the vector into the genome. When the cell state was stable, blasticidin was added to the medium for positive cell selection. During the selection process, negative cells will die, and when the cell state fully recovers to normal, it indicates the completion of the construction of the BmE-sgRNA cell line. BmE-sgRNA cells were seeded in six-well plates with a density of 70–90%, and transfection was performed approximately 24 h later. Plasmids (4.0 μg) were transfected into BmE-sgRNA cells using X-tremeGENE HP DNA transfection reagent (Roche) (Basel, Switzerland). All transfections were carried out according to the manufacturer’s instructions.

### 2.3. Cas9 Plasmid Transfection

The constructed cell lines were evenly distributed into six-well cell plates, with three wells per group and a total of three groups. Each group was transfected with Cas9, xCas9 3.7, and Cas9-NG plasmids, with 3.2 μg of plasmid and 6 μL of Roche transfection reagent used for transfection. Cells were collected for genomic extraction 7 days after transfection.

### 2.4. Mutation Detection and Analysis

The primer design utilized the PCR amplification of flanking sequences at both ends of the target locus, resulting in a singular sequencing product that was submitted for high-throughput analysis. The analysis of deep sequencing data involved custom shell scripts, with initial preprocessing conducted using Trimmomatic (version 0.39) software (England) for quality control and data filtering. Paired-end reads were merged using FlASH software (version 1.2.11) (USA), and non-target sequences were removed using Cutadapt software (version 1.18) (Sweden). Next, the editing results of the target sequences were analyzed using CRISPResso (version 2.0.45) (USA).

## 3. Results

### 3.1. A Library Approach to Evaluate the Function of Cas9 Variants in High-Throughput

The editing efficiency with Cas9 and its variants is affected by the variations themselves and by differences in sgRNA sequences and species. Therefore, the optimization of the nucleotide sequences of Cas9 variants suitable for various target species is crucial for achieving the highest expression and efficiency when evaluating the practicality of Cas9 variations. First, given that various codon optimizations can influence Cas9 protein expression, equivalent codon optimizations for Cas9, xCas9-3.7, and SpCas9-NG in *B. mori* were utilized, except for the seven amino acids in both the xCas9-3.7 and SpCas9-NG sequences, which differed from those in Cas9 (Figure 1B). Variant codons were selected based on GenScript recommendations. The Cas9 portion of the previously validated expression system [40] was replaced with the optimized Cas9 and its variants, and the IE2 promoter was uniformly substituted to amplify protein expression and enhance editing efficiency (Figure 1A). Second, a maximum number of sgRNAs within the same PAM sites were designed to compensate for either the inefficiency of certain sgRNAs during editing or the editing impact caused by the inability to edit the target sites. To achieve this, a large number of sgRNA vectors were designed, synthesized, and constructed and all the target sites were tested after editing. Additionally, it is necessary to design and synthesize a substantial number of PCR primers. This process requires substantial investments in both time and money. This requires significant financial and time resources. In addition, despite dedicated efforts, the editing ability of Cas9 variants may not be fully reflected. A comparable situation was encountered when evaluating the Cas9-variant editing system in *B. mori*, expending a significant amount of time and producing unsatisfactory outcomes. To overcome the above limitations, our envisioned development includes a system that only requires one sgRNA to detect all potential PAM sites while allowing the high-throughput sequencing of all targets in one step. This restored the Cas9 variant’s true editing ability to the maximum possible extent.

Based on the aforementioned considerations, an sgRNA sequence with no target region in the silkworm genome was designed. Next, the sgRNA scaffold sequence was included following the sgRNA sequence; simultaneously, a homologous sequence of the sgRNA was developed as a target sequence to be added after the scaffold (Figure 1C). For the PAM sequence, an arbitrary combination of 3 base pairs (4 × 4 × 4) was added following the target sequence, resulting in 64 sequences (Figure 1C). The designed sequences were assembled into a pBac transgenic vector containing a U6 promoter. This enables the vector to transcribe sgRNA normally, which can subsequently be targeted at the vector itself when Cas9 variants and vectors coexist, and the variants recognize the PAM sequences on the vectors (Figure 1C). A single pair of detection primers was sufficient to detect all target sites since all target sequences were identical except for the PAM sequence. The corresponding editing outcomes were detected while verifying that the Cas9 variants recognized the PAM sequences (Figure 1C). A zeocin-resistance marker was added to the sgRNA vector and screened for zeocin, and a stable cell line expressing sgRNA was generated. Using this system, 64 PAM sites were simultaneously detected by transfecting any vector of the Cas9 variants (Figure 1C). Subsequently, the recognizable PAM sites were modified, and the target site sequences were amplified using only one pair of PCR primers. The recognizable PAM sequences of the *B. mori* variants were established through high-throughput sequencing and information analysis. The validation period for Cas9 variants in *B. mori* can be significantly reduced by eliminating the effects of sgRNAs and the species themselves. This process maximizes the restoration of Cas9-variant editing potential and accelerates the expansion of CRISPR-based insect genome-editing tools. In addition to its application to silkworm cells, this system can serve as a reference for extending the CRISPR editing system to other insects. Furthermore, it can be utilized in mammalian cells to verify the editing capabilities of newly developed Cas9 variants.

### 3.2. High Throughput Screening to Determine the Editing Ability of xCas9 3.7 in BmEs

xCas9 3.7 was derived through the directed evolution of Cas9. Researchers have employed a phage-assisted continuous evolution (PACE) system, which allows for the rapid generation of Cas9 variants that accept a wider range of PAM sequences. Twenty-six Cas9 versions were produced with varying efficiencies and PAM sequences, from which xCas9 3.7 was ultimately chosen for its high efficiency in recognizing PAM sequences such as NG, NNG, GAA, GAT, and CAA [37]. These results were obtained using the PACE system, which was developed to identify bacteria. However, in subsequent practical applications, not all PAMs were recognized. Based on similar studies, the editing results of the NG PAM sequences were tested.

xCas9-3.7 and wild-type Cas9 (Cas9-WT) were chosen based on their efficiencies reported in the literature and transferred into our newly constructed high-throughput screening platform. Cells were collected 7 days after transfection to extract the genome and amplify the target sites for deep sequencing. After analyzing the results, the mutation rate of each PAM was measured. Cas9-WT displayed high editing proficiency at all standard PAM sites, whereas xCas9 3.7 exhibited a comparatively lower efficiency than Cas9-WT across all four sites. In particular, xCas9 3.7 had significantly less editing effectiveness in CGG PAMs (Figure 2A). At the same time, at the AGG and TGG sites, the editing efficiencies were low (4.05% and 2.62%, respectively). xCas9 3.7 displays an editing efficiency comparable to that of Cas9-WT only at the GGG site (Figure 2A). This phenomenon is consistent with the results of previous experiments conducted on human cells [37]. The sequencing data was analyzed for the remaining 12 atypical PAMs, and it was discovered that Cas9-WT had no noteworthy editing capacity at any of these locations, which was in line with expectations, given the usual untargetable nature of these loci by Cas9-WT (Figure 2A). Compared to Cas9-WT, xCas9 3.7 exhibited editing activity at all 12 loci. Analysis of the atypical loci indicated that the editing efficiency of CGC was the lowest (0.12%), whereas that of TGC was the highest (59.21%) (Figure 2A). When the PAM sequence was AGA, the editing efficiency was 1.71%. The editing efficiency was 6.40% when the PAM site was TGA, and the editing efficiency did not vary significantly when the PAM sites were CGA and GGA, which were 2.99% and 2.91%, respectively (Figure 2A). Overall, the editing efficiency of xCas9 3.7 was low when the PAM sequence was NGA. When the PAM sites TGT and GGT were present, xCas9 3.7 demonstrated an editing ability similar to that of Cas9-WT, achieving efficiencies of 20.25% and 19.86%, respectively. xCas9 3.7 also displayed some degree of editing activity on the remaining PAM sites of AGT and CGT, with editing efficiencies of 2.27% and 7.42%, respectively. For the four NGC sites, excluding TGC, which exhibited the highest editing efficiency of 59.21%, the editing efficiencies of the other three sites (AGC, CGC, and GGC) were generally low at 4.35%, 0.12%, and 1.38%, respectively (Figure 2A). Overall, xCas9 3.7 demonstrated high editing efficiency in all atypical *B. mori* PAM loci, specifically in the TGT, TGC, and GGT loci. Among these motifs, the efficiency of editing with xCas9 3.7 is comparable to or greater than that of Cas9-WT at the NGG locus. Particularly, the editing efficiency of the TGC motif approached 60%. Although other loci also exhibited some editing efficiency, their efficiency was not as high as those at the TGC locus. These results suggest the potential of xCas9 3.7 for gene editing applications in *B. mori*. This editing ability differed slightly from that of human cells. It is understandable that there is a preference for xCas9 3.7, which recognizes PAM sites in different species, and that there is also variability in editing efficiency.

### 3.3. High Throughput Screening to Determine the Editing Ability of SpCas9-NG in BmEs

SpCas9-NG has been identified as a superior Cas9 variant to xCas9 3.7 for targeting non-canonical NG PAMs. Unlike xCas9 3.7, which is a randomly mutated version of the entire Cas9 protein sequence aimed at obtaining Cas9 variants possessing non-site-specific editing capabilities, SpCas9-NG uses a distinct method, starting with Arg1333 and Arg1335 in the PAM recognition region PI of the Cas9 protein itself [36]. These regions were used to identify the second and third G bases in the Cas9 PAM sequence. PAM restriction to the third base of NGG was reduced by modifying Arg1335. Mutations were introduced at other sites to compensate for the loss of specific interactions. A series of amino acid combination experiments led to the selection of the optimal combination: R1335V/L1111R/D1135V/G1218R/E1219F/A1322R/T1337R (Figure 1B). This resulted in a breakthrough in the NGG PAM restriction. This combination was termed SpCas9-NG. Due to the distinct modification technique employed with xCas9 3.7, SpCas9-NG recognizes a PAM sequence that differs slightly from that identified by xCas9 3.7. Compared to Cas9, SpCas9-NG displayed increased activity on NGH PAM (H for A, T, or C) and showed some editing efficiency on NGG PAM [36]. Furthermore, we created an SpCas9-NG protein specifically for *B. mori* and subsequently transferred it to a cell bank. After 7 days, the cells were harvested, and their genomes were extracted. The target and PAM sequences were amplified, and high-throughput sequencing was performed.

After analyzing the results, the mutation percentage for each PAM was determined. The analysis revealed that Cas9-WT displayed high editing ability at all four typical PAM loci. Notably, SpCas9-NG demonstrated superior performance at all four loci, particularly at the TGG locus, where it achieved a higher editing activity of 23.98% compared to Cas9-WT. At three other loci (AGG, CGG, and GGG), SpCas9-NG displayed editing efficiencies of 11.83%, 11.08%, and 6.81%, respectively (Figure 2B). Further analysis of the sequencing results from atypical PAM loci showed that AGT, CGA, and TGC had similar editing efficiencies to Cas9-WT at the NGG locus, at 28.7%, 24.6%, and 51.07%, respectively (Figure 2B). Additionally, editing activity was observed for AGC, CGT, GGT, and TGA with editing efficiencies of 7.58%, 5.73%, 5.59%, and 9.9%, respectively. The editing activity of SpCas9-NG at the AGA (0.4%), GGA (4.46%), TGT (4.68%), CGC (1.74%), and GGC (0.46%) sites was negligible (Figure 2B). Similarly, SpCas9-NG demonstrated the ability to modify unconventional PAM sites in BmE cells to some degree. Editing efficiencies of over 20% were observed at the CGA, AGT, and TGC sites, whereas the efficiencies at AGA (0.4%) and GGA (1.46%) were low. The editing ability was demonstrated to be less than 5% in TGT (4.68%), CGC (1.74%), and GGC (0.46%) (Figure 2B). Overall, these results suggested that SpCas9-NG possesses editing capabilities beyond those of conventional PAM sites. The potential of SpCas9-NG for gene editing applications in BmE cells of the *B. mori* was shown. However, the editing ability in this case is slightly different from that of human cells. This disparity can be explained by the preference for PAM sites identified by SpCas9-NG, which varies across different species.

### 3.4. Genome-Wide Analysis of Targetable Sites

A lateral comparison of the editing proficiency between xCas9 3.7 and SpCas9-NG in BmE cells was conducted. Initially, the editing efficacy of typical NGG PAM sites where xCas9 3.7 exhibited superiority was compared to SpCas9-NG at the GGG site. In addition, SpCas9-NG proved to be much more efficient than xCas9 3.7 at the other three sites (Figure 3B). Furthermore, among the 12 atypical PAM loci, SpCas9-NG exhibited higher editing proficiency than xCas9 3.7 at the TGA, CGA, AGT, and AGC loci. Compared to SpCas9-NG, xCas9 3.7 demonstrated superiority over SpCas9-NG at the AGA, GGA, TGT, and GGT loci. Both proteins showed comparable editing abilities at the CGT and TGC sites, with more than 50% efficiency, particularly at the TGC site (Figure 3B). Although each protein has its advantages and disadvantages, their editing efficiencies did not exceed 2% at the CGC and GGC sites. In addition to the aforementioned two loci, in the remaining 14 loci, both xCas9 3.7 and SpCas9-NG possess editing capabilities, with a slight preference. When these two proteins are combined, we can effectively target most PAM loci of NGN found in *B. mori* (except CGC and GGC, actual efficiency may not be high based on sequencing results).

After confirming the efficiency of editing the two Cas9 variants in BmE cells, a horizontal assessment of individual genes was conducted to identify possible sgRNA targets of the three proteins. The *BmFib-H* gene was chosen, which is economically significant in *B. mori*, to analyze the number and distribution of recognizable sites for SpCas9, xCas9 3.7, and SpCas9-NG. Our findings showed 2736 SpCas9 recognizable sites throughout the *BmFib-H* gene, including the promoter, exon, and intron segments. xCas9 3.7 and SpCas9-NG yielded 7503 and 8395 editing sites, respectively (Figure 3A). The combined application of the three proteins increased the number of targetable sites 6.8-fold. Furthermore, in addition to single genes, a comprehensive genome-wide bioinformatics analysis was conducted to identify all possible sgRNA targets. The densitometric analysis demonstrated the uniform distribution and coverage of all loci in the genome. The analysis identified 33,819,862 sites recognized by Cas9; 129,885,250 sites recognized by xCas9 3.7; and 121,021,823 sites recognized by SpCas9-NG. The range of editable segments was expanded by 3.8 and 3.5 times, respectively (Figure 3C). The Cas9 variant showed PAM sites every 13 base pairs in the genome, with an average distance between 2 specific targeting sites of 3.4 base pairs for xCas9 3.7 and 3.6 base pairs for SpCas9-NG in comparison to Cas9. According to the genome-wide distribution analysis displayed in Figure 3C, all targetable sites were equally distributed throughout the genome. In conclusion, the aforementioned integrated system significantly increases the number of available high-resolution target locations and guarantees efficient gene manipulation at the most essential positions.

### 3.5. Analysis of Deletion Patterns in Three Cas9 Proteins

Although both xCas9 3.7 and SpCas9-NG are mutations in Cas9, a comparative analysis indicated that they exhibit distinct efficiency levels. Data analysis revealed that the deletion patterns of the three proteins varied slightly. Therefore, it was necessary to compare the editing results of both methods and to determine whether they were similar to the deletion patterns produced by Cas9 editing. In previous research, it has been found that deletions are responsible for the majority of mutations, ranging from 61.50% in *M. musculus* to 73.11% in Homo sapiens [41]. Therefore, this study primarily compared the differences in deletion patterns among the three species. Mutations resulting in deletions were classified into two types: uniquely oriented deletions, which refer to mutations in which the deletion only occurs on one side of the cleavage site, and bidirectional deletions, which refer to mutations in which the deletion occurs at both ends of the cleavage site. In a previous study, researchers revealed that the dominant form of deletion in *B. mori* was a uniquely oriented deletion consisting primarily of small-fragment deletions [41]. This phenomenon was mainly observed upstream of the cleavage site, unlike in other species.

The editing outcomes of two PAM sites, CGG and GGG, in the Cas9-WT results were randomly analyzed. The analysis showed that Cas9-WT primarily caused the deletion of one or two bases located upstream of the PAM site in a single direction. This result is consistent with the previous editing results for *B. mori* (Figure 4A,D). The deletion pattern of xCas9 3.7 was analyzed by randomly selecting an atypical PAM site (TGC) and a typical PAM site (AGG). The small-fragment deletion pattern was still dominant, and the deletion position was located upstream of the PAM site, which is consistent with the results for Cas9-WT. However, xCas9 3.7 showed more base deletions at the atypical site (Figure 4B,E). Similarly, the deletion patterns of SpCas9-NG were analyzed using both typical and atypical PAM sites (TGG and TGA). Similar to the xCas9 3.7 editing results, SpCas9-NG maintained the same deletion pattern as Cas9-WT. However, the deleted fragments in SpCas9-NG were longer than those in Cas9-WT (Figure 4C,F). To determine if this phenomenon is an exception or a general trend, we tallied the editing results of Cas9, xCas9 3.7, and SpCas9-NG separately. The results showed that all three proteins mainly caused deletions of 1bp (Figure 4G), with little difference in the proportion of deletions of long fragments. Thus, this situation may be an exception, and the primary editing patterns of xCas9 3.7 and SpCas9-NG continue the deletion pattern of Cas9. 

The amino acid sites altered in both variants did not involve the two structural domains responsible for cleavage—Ruvc and HNH—and maintained the same cleavage pattern as Cas9. Therefore, they maintained the same cleavage mode as Cas9 did. This explains why the deletion sites of the three Cas9 proteins are located upstream of the PAM site. Deletion patterns also remained largely unchanged. Statistical analysis of the editing results showed that xCas9 3.7 and SpCas9-NG had a deletion pattern similar to that of Cas9.

## 4. Discussion

The power and impact of genome editing have been demonstrated in cultured cell lines and individuals of various organisms, including microbes, plants, animals, and even human embryos. As an economically important insect and research model for lepidopterans, *B. mori* lacked efficient site-specific gene-manipulation tools until the advent of genome-editing tools. Gene-function research in *B. mori* has undergone significant changes through the use of genome-editing technologies, including ZEN, TALEN, and CRISPR/Cas9. Despite the progress made, limitations in Cas9 persist, preventing the accurate editing of certain loci. Therefore, it was necessary to edit a broad range of Cas9 variants. 

The use of CRISPR/Cas9-based editing systems has been severely limited due to PAM sequence constraints. Several approaches have been proposed to address this issue. Kleinstiver devised a PAM recognition domain based on the structural information of the Cas9 protein to identify NAG and NGA, in addition to canonical NGG [42]. Bolukbasi et al. fused another DNA-binding domain to Cas9 and broadened PAM to NAG and NGC [43]. Other researchers constructed and employed CRISPR proteins that possess distinct PAMs derived from *Neisseria meningitidis* [44], *Streptococcus thermophilus* [42], and *Staphylococcus aureus* [30]. Researchers have also developed novel genome-editing tools. 

As research on genome-editing technology continues to advance, there is a need for proteins that can achieve a wider range of editing. To address this issue, Hu et al. targeted the evolution of the Cas9 protein and screened successive variants until one with an improved editing ability was found [37]. In the same year, Nishimasu et al. targeted the PI region of the PAM loci [36]. By creating a series of amino acid substitutions, they expanded the range of gene editing. The two aforementioned variations of Cas9 retain the advantages of the original Cas9, thereby increasing its generalizability. In this study, we present two powerful Cas9 variants based on the CRISPR/Cas9 genome-editing platform, both of which have the ability to edit atypical PAM sites outside the NGG. This study found that SpCas9-NG and xCas9 3.7 possess unique beneficial sites, with SpCas9-NG having a slightly higher overall efficiency than xCas9 3.7. xCas9 3.7 showed a lower editing efficiency than Cas9 at the NGG locus; however, it exhibited a higher editing efficiency at the atypical PAM loci TGT, TGC, and GGT than Cas9 at the NGG PAM locus. xCas9 3.7 also displayed editing efficiency at the AGC, TGA, and CGT loci. Unlike xCas9 3.7, SpCas9-NG exhibited editing efficiency comparable to that of Cas9 at the NGG PAM site, and it also displayed similar or even higher editing efficiency than Cas9 at the NGG PAM site on atypical PAM sites such as AGT, TGC, and CGA. It displayed editing efficiency at the AGC, TGA, TGT, CGT, and GGT locations. These two proteins exhibited dissimilar preferences for unusual PAM sites. By combining these phenomena, we can effectively modify the NG PAM locus in the genome of *B. mori* by utilizing these three proteins in a flexible manner that meets the specific needs of the locus. 

xCas9 3.7 and Cas9-NG are mature Cas9 variants that have been widely used in plants and mammals [45,46], but they exhibit significant differences in editing efficiency in different species. Unlike in humans, where the editing results are more efficient, in plants, the recognized PAM sequences are fewer and the efficiency is lower [45,47]. Similarly, research on xCas9 3.7 in fruit flies has shown that, besides NGG, it can only recognize one PAM sequence (TGA) [48]. This phenomenon is complex and involves differences in the editing efficiency of Cas9 in different species, such as potentially lower editing efficiency of Cas9 in fruit flies. It is also influenced by the sgRNA sequence, as different sgRNAs with the same PAM sequence can have varying editing efficiencies [3]. This study overcomes these limitations by using sgRNA sequences with neutral editing efficiency, which can better approximate the true editing efficiency of Cas9 in the species being studied. Therefore, in this study, it was observed that Cas9 identified more PAM sites.

In conclusion, we introduced a platform that validated Cas9-variant editing activity in BmE cells. This platform confirmed the recognizable PAM sites of xCas9 3.7 and SpCas9-NG variants while exhibiting the ability to validate the viability of Cas9 variants quickly and easily. This study presents new proteins that have demonstrated high efficiency and wide editing capabilities for exploring gene function and performing saturation editing in silkworms. These proteins offer new possibilities for editing silkworm genomes at specific sites. Given advancements in base editing technology and the growing demand for saturation editing, Cas9 variants with a wider editing range are becoming increasingly necessary. These proteins also hold significant value for functional gene research and the rapid breeding of economically important varieties of silkworms. The high-throughput platform for detecting Cas9 variants designed and constructed in this study can quickly perform numerous feasibility assays for Cas9 variants. Moreover, it can aid in the creation of efficient genome-editing platforms that are useful for other insects, particularly those lacking NGG PAM sequences.

## Figures and Tables

**Figure 1 insects-15-00241-f001:**
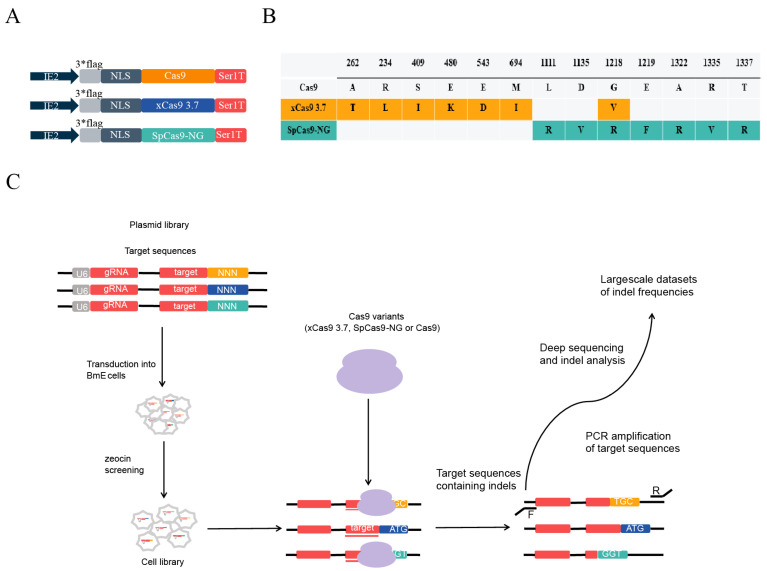
Establishing a High-Throughput Screening System (**A**): This study used Cas9 and its variants, represented schematically by vectors. IE2 promoter; NLS, nuclear localization signal; ser1T, Sericin 1 Terminator. (**B**) Genotypes of the engineered Cas9 variants. The changed amino acid residues are colored. (**C**) Schematic representation of high-throughput assessment of xCas9, SpCas9-NG and Cas9 activity. A plasmid library containing a pair of guide RNA coding sequences and their corresponding target sequences was transduced into BmE cells to generate a cell library. Subsequently, xCas9, SpCas9-NG, or Cas9 was introduced into these cell libraries to induce indels on the integrated target sequences. PCR amplification and deep sequencing of the integrated target sequences were performed to measure indole frequency.

**Figure 2 insects-15-00241-f002:**
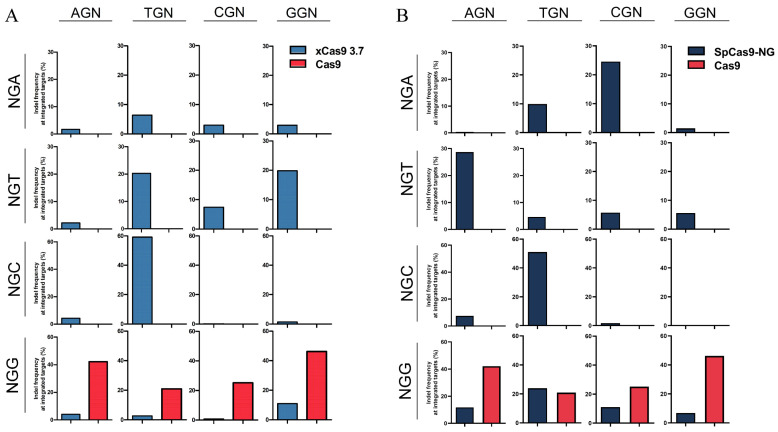
Assessment of the targeting efficiency, editing efficiency, and targeting specificity of xCas9 3.7 and SpCas9-NG in BmE cells. (**A**): Genome editing at 16 NGN PAM sites by Cas9-WT and xCas9 3.7. The target sites covered all 16 possible NGN PAM combinations. (**B**): Genome editing at 16 NGN PAM sites by Cas9-WT and SpCas9-NG. The target sites covered all 16 possible NGN PAM combinations.

**Figure 3 insects-15-00241-f003:**
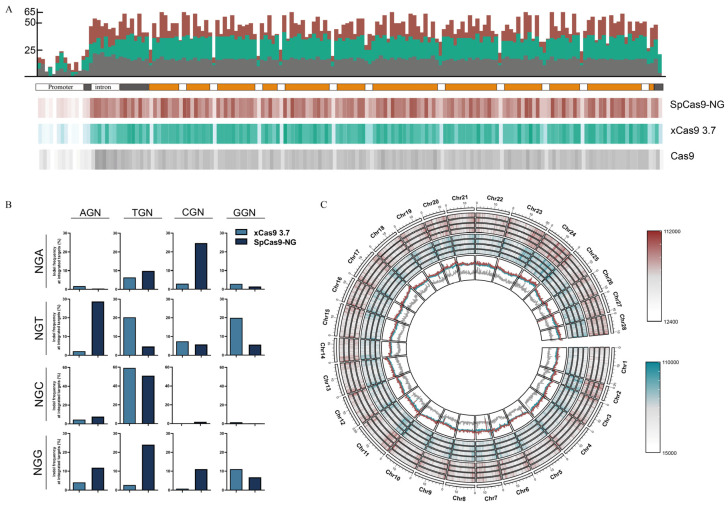
Genome-wide analysis of targetable sites (**A**): Numbers and distributions of sgRNA sites along *BmFib-H*. Top column is the number of target sites. *X*-axis represents the position on *BmFib-H*, and *Y*-axis represents number of target sites on each 100 bp window. Middle column is the structure of *BmFib-H*. Bottom column is the site’s distribution. Each line indicates a target site. (**B**): Comparison of the efficiency of both xCas9 3.7 and SpCas9-NG at 16 Pam sites. (**C**): Genome-wide distribution of targetable sites. Curry for SpCas9-NG, green for xCas9 3.7.

**Figure 4 insects-15-00241-f004:**
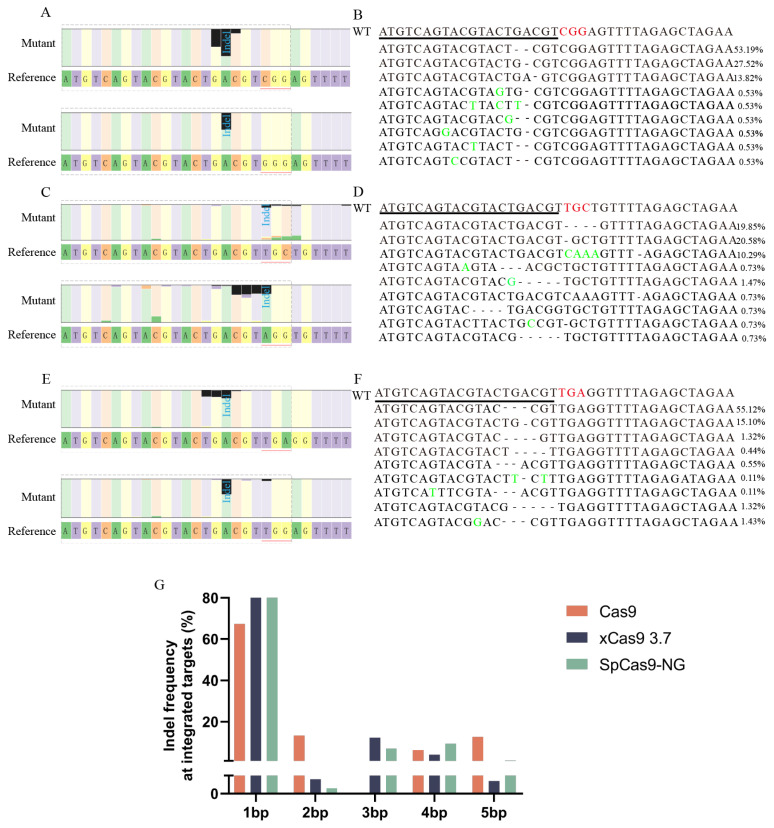
The deletion patterns of Cas9, xCas9 3.7 and SpCas9- NG (**A**) The deletion patterns of Cas9. (**B**) The deletion patterns of xCas9 3.7. (**C**) The deletion patterns of SpCas9-NG. (**D**) Sequence information after Cas9 editing when PAM site at CGG. (**E**) Sequence information after xCas9 3.7 editing when PAM site at TGC. (**F**) Sequence information after SpCas9-NG editing when PAM site at TGA. (**G**) Deletion length statistics following the editing of three Cas9 proteins.

## Data Availability

All raw sequencing data are available from the NCBI database with BioProject accession PRJNA1061364. The corresponding author can provide all the created assemblies for this work upon request.

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
