# Peer review of "High-Throughput Screening of PAM-Flexible Cas9 Variants for Expanded Genome Editing in the Silkworm (Bombyx mori)"

_insects, 2024, doi:10.3390/insects15040241_

Round 1

Reviewer 1 Report

Comments and Suggestions for Authors

Sun et al. established a novel Cas9 variant detection platform in their manuscript. They successfully verified the editing ability of xCas9 3.7 and Cas9-NG in the Bombyx mori (B. mori). It was observed that the two proteins have different recognition sites in the B. mori, which complement each other and greatly expand the editing ability of the CRISPR/Cas9 editing system in the B. mori. At the same time, it is evident that this detection platform is not restricted to the B. mori system but can also be implemented in other insects, indicating great potential for expansion. The manuscript is well-written and easy to comprehend. However, there are still some issues that need to be addressed:

1.     Both Cas9-NG and SpCas9-NG expressions are present in the article. It is recommended to standardize and choose one expression.

2.     To ensure the reader understands the cell engineering operations involved in the experiments, it is recommended to provide a clear and concise description of the application of zeocin resistance markers, the screening process, and the establishment of stable cell lines.

3.     The article mentions that the inspection platform has the potential to be extended to mammals and other insects. However, it is unclear why it has this potential for other species.

4.     The editing efficiency of xCas9 3.7 and Cas9-NG is not high at certain sites. In the results section of 3.4, are the recognizable PAM sites theoretically editable sites or are they statistically editable sites based on the experimental results of obtaining specific recognized sites?

5.     Cas9 editing triggers a repair process that not only results in deletions, but also introduces substitutions. The article primarily focuses on analyzing deletions. Why was the proportion of base substitutions not analyzed?

6.     Regarding the introductory section, please consider adding additional references.

7.     line 169: "temporal" in "significant financial and temporal resources" may not be appropriate, suggest amending to "time".

8.     line 210: "mammalian cells to rapidly verify the editing capabilities of newly developed Cas9 variants" This sentence is a bit redundant and could be simplified to "mammalian cells to verify the editing capabilities of newly developed Cas9 variants."

9.     Line 223: "efficiencies in literature and transferred" - It is recommended to correct to "efficiencies reported in the literature and transferred"

Comments on the Quality of English Language

Minor editing of English language required.

Author Response

Thank you for your recognition of our study. It is our honor to have our paper reviewed and commented by you. Your profound insights will improve the quality of our paper. In accordance with your suggestions and comments, we have revised and corrected the manuscript. Please see the attachment for specific details.

Reviewer 2 Report

Comments and Suggestions for Authors

xCas9 3.7 and SpCas9-NG are valuable genome editing tools. xCas9 3.7 and SpCas9-NG were introduced into the silkworm genome editing work, which expanded the toolbox of silkworm genome editing and helped to improve the region and quality of silkworm genome editing. There are still some problems to be solved in this paper.

1.     Material and method part is too rough, many details missing, need to be added.

2.     In the main text, various terms are used to refer to the Cas9 employed for the control group, such as Cas9, spCas9, Cas9 - WT. Whether these refer to the same Cas9 gene, if so, it is best to have a unified name.

3.     The editing efficiency in figure 2 didn’t provide the variance data, how many samples were used in investigation? Single sample was no statistically significant.

4.     Y-axis of Figure 4 A - C exist issues, both of mutant and reference sequences were marked as reference, need to correct.

5.     The discussion section did not provide an analysis based on the current result. Extensive investigations on the targeting efficiency and editing efficiency of xCas9 3.7 and SpCas9-NG have already been conducted in other species. I suggested that the authors compare their results with other studies in the discussion section and discuss the reasons for observed differences.

For example:

Kim H K, Lee S, Kim Y, et al. High-throughput analysis of the activities of xCas9, SpCas9-NG and SpCas9 at matched and mismatched target sequences in human cells[J]. Nature biomedical engineering, 2020, 4(1): 111-124.

Hua K, Tao X, Han P, et al. Genome engineering in rice using Cas9 variants that recognize NG PAM sequences[J]. Molecular Plant, 2019, 12(7): 1003-1014.

Author Response

(The authors gave the same response as above.)

Round 2

Reviewer 2 Report

Comments and Suggestions for Authors

The author has revised most of my concerns except for one.

The text of Y-axis of Fig. 4 A-C, the sequences that contain mutant sites are also marked as 'Reference'. It should be 'Mutant' not 'Reference'. Only the sequence at the bottom of subgraph A-C is the reference sequence.

Author Response

Thank you very much for your suggestion, we have made the modifications. Please see the attachment.
